# OpenShape: Scaling Up 3D Shape Representation Towards Open-World Understanding

Minghua Liu[1][*]     Ruoxi Shi[2][*]     Kaiming Kuang[1][*]     Yinhao Zhu[3]     Xuanlin Li[1]

Shizhong Han[3]     Hong Cai[3]     Fatih Porikli[3]

Hao Su[1]

[1] UC San Diego     [2] Shanghai Jiao Tong University     [3] Qualcomm AI Research[†]

Project Website: `https://colin97.github.io/OpenShape/`

## Abstract

We introduce OpenShape, a method for learning multi-modal joint representations of text, image, and point clouds. We adopt the commonly used multi-modal contrastive learning framework for representation alignment, but with a specific focus on scaling up 3D representations to enable open-world 3D shape understanding. To achieve this, we scale up training data by ensembling multiple 3D datasets and propose several strategies to automatically filter and enrich noisy text descriptions. We also explore and compare strategies for scaling 3D backbone networks and introduce a novel hard negative mining module for more efficient training. We evaluate OpenShape on zero-shot 3D classification benchmarks and demonstrate its superior capabilities for open-world recognition. Specifically, OpenShape achieves a zero-shot accuracy of $46.8\%$ on the 1,156-category Objaverse-LVIS benchmark, compared to less than $10\%$ for existing methods. OpenShape also achieves an accuracy of $85.3\%$ on ModelNet40, outperforming previous zero-shot baseline methods by $20\%$ and performing on par with some fully-supervised methods. Furthermore, we show that our learned embeddings encode a wide range of visual and semantic concepts (e.g., subcategories, color, shape, style) and facilitate fine-grained text-3D and image-3D interactions. Due to their alignment with CLIP embeddings, our learned shape representations can also be integrated with off-the-shelf CLIP-based models for various applications, such as point cloud captioning and point cloud-conditioned image generation.

## 1   Introduction

3D shape understanding has recently garnered a surge of interest driven by the growing demands in real-world applications, such as augmented/virtual reality, autonomous driving, and robotics. Despite significant advancements in 3D recognition and analysis, existing data-driven approaches are still greatly limited by the scale of 3D training datasets and tend to exhibit poor generalization when facing unseen shape categories, hindering the deployment of existing models in real-world applications.

Note that 3D shapes and 2D images can be easily linked through rendering, and the dataset scale issue of 2D images has been remarkably addressed, as shown in recent works such as CLIP [55]. Therefore, many recent studies aim to utilize pre-trained 2D image-language models [55, 59] to assist 3D tasks, such as 3D generation [22, 26, 45, 63, 33, 7, 38, 67] and 3D scene-level segmentation [18, 27, 14, 79,

---

[*]Equal Contribution

[†]Qualcomm AI Research is an initiative of Qualcomm Technologies, Inc.

37th Conference on Neural Information Processing Systems (NeurIPS 2023).

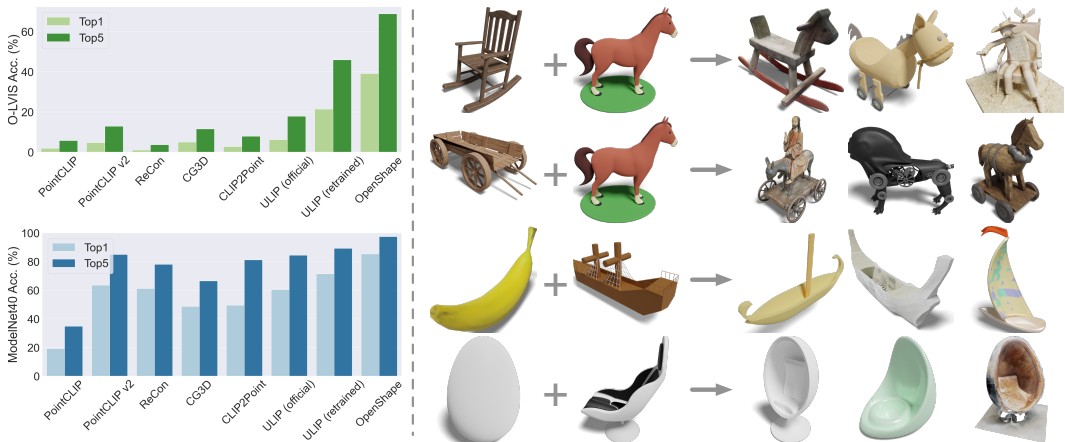

Figure 1: **Left**: Zero-shot shape classification on the Objaverse-LVIS (1,156 categories) and Model-Net40 datasets. OpenShape outperforms previous methods by a large margin. We exclude shapes in Objaverse-LVIS during training, and we also retrain ULIP [78] on our ensembled training shapes for fair comparison. **Right**: Our shape representations encode a broad range of semantic and visual concepts. We input two 3D shapes and use their shape embeddings to retrieve the top three shapes whose embeddings are simultaneously closest to both inputs. See Section. 4.4 for more details.

41, 49]. Regarding 3D shape-level understanding, a straightforward idea is to project 3D data to the 2D domain through rendering and use CLIP to analyze the 2D images, thereby enabling zero-shot 3D shape classification [86, 88]. However, these methods suffer from occlusion and information loss during projection, and unnecessary latency due to point cloud rendering and multiple CLIP inferences.

To overcome the limitations caused by projection, it is necessary to train a 3D-native model by distilling knowledge from pretrained 2D models. However, training a 3D-native model requires a set of 3D shapes, and the amount of knowledge that can be distilled is determined by the size of the 3D dataset. For example, ULIP [78] aims to learn a joint representation space between language, 2D images, and 3D shapes, but uses a small-scale 3D dataset ShapeNetCore [8] for knowledge distillation. Specifically, ULIP fixes the 2D CLIP text and image encoders and trains a dedicated 3D-native point cloud encoder to extract 3D shape representations. The 3D encoder strives to align the 3D shape embedding space with the CLIP image and language embedding spaces by utilizing contrastive learning across all three modalities. However, since ULIP is only trained on 52K shapes of 55 object categories, it still struggles with out-of-distribution shape categories and fails to demonstrate an impressive open-world understanding of 3D shapes.

In this work, we propose a novel method called OpenShape, which follows a similar paradigm as ULIP but aims to achieve a more generalized and scalable joint representation space encompassing language, 2D images, and 3D shapes. Our focus mainly lies on scaling up representation learning and addressing corresponding challenges. In OpenShape, we emphasize four key factors during the training process: (a) data scale: we significantly increase the scale of 3D training data by combining four public 3D shape datasets, resulting in 876k 3D shapes covering much more diverse categories; (b) text quality: the 3D shapes from our main dataset, Objaverse [12], is dominated with inaccurate or uninformative text descriptions. Given the data scale, we propose three strategies to automatically filter and enrich the text descriptions; (c) 3D backbone scaling: since most existing 3D backbones target small datasets, we find that it's important but non-trivial to scale up the 3D backbones; and (d) data resampling: since the ensembled dataset is highly unbalanced, we utilize hard negative mining to improve the model's discriminative ability.

We first evaluate OpenShape on the zero-shot 3D shape classification task. As shown in Figure 1, OpenShape outperforms previous zero-shot approaches on the ModelNet40 dataset by at least 20%. Moreover, OpenShape excels at handling long-tail categories. On the challenging Objaverse-LVIS dataset, which contains 1,156 categories, OpenShape achieves a 46.8% accuracy, significantly surpassing previous methods. Notably, this performance gap remains even when ULIP is retrained on our ensembled datasets, highlighting the superiority of our text enrichment and training strategies.

Besides zero-shot classification, we present demos that showcase the wide range of visual and semantic concepts learned by OpenShape. For example, in Figure 1-right, we take two 3D shapes as input and use their OpenShape embeddings to retrieve the top three shapes whose embeddings are simultaneously closest to both inputs from our ensembled dataset. The retrieved shapes exhibit an interesting combination of the semantic and geometric elements from both input shapes. Furthermore, since we align our 3D shape embedding space with the CLIP language and image embedding space, we demonstrate that OpenShape embeddings can be easily integrated with other CLIP-based models to perform cross-modality tasks such as point cloud captioning and point cloud-conditioned image generation.

## 2 Related Work

### 2.1 CLIP for 3D Learning

Image-language models like CLIP have achieved remarkable performance through large-scale image-text pretraining [55, 29, 35, 84, 4, 56, 61, 36]. As these models excel at capturing rich visual concepts and possess impressive zero-shot capabilities, they have been applied to various 3D vision tasks. For instance, numerous recent works utilize CLIP to facilitate zero-shot text-to-3D generation [22, 26, 45, 63, 33, 7, 32, 5, 28, 77, 40, 83], typically through CLIP-guided per-scene optimization. From a recognition perspective, some works focus on scene-level representation, aiming to leverage CLIP priors for zero-shot 3D segmentation or detection in both indoor [18, 27, 14, 79, 41, 49, 82, 23, 60, 85, 31] and outdoor scenes [9, 21]. Meanwhile, another line of work focuses on shape-level understanding, targeting zero-shot shape classification [86, 88, 53, 78, 19] and part segmentation [39, 1]. There are two primary working paradigms for these methods. The first [86, 88, 24] involves using images as a medium representation, projecting 3D point clouds into 2D and employing 2D CLIP for inference. However, these methods typically suffer from occlusion and information loss during projection, along with unnecessary latency due to point cloud rendering and multiple 2D CLIP inferences. The second paradigm involves training a 3D-native encoder attempting to distill or fuse CLIP features into 3D representations. Our paper follows this paradigm.

### 2.2 3D Shape Representation Learning

Various works have studied self-supervised pretraining for point clouds by designing pretext tasks [15, 69, 50, 2, 66] such as self-reconstruction [57, 13, 3, 72], masked auto-encoding [48, 80, 20], distortion reconstruction [64, 44, 68], normal estimation [57], and contrastive learning [87, 62, 76]. These tasks enhance models' shape representations and improve their performance on downstream applications, although they do not involve multimodal semantic alignments during pretraining.

Recently, some works [53, 78, 19], exemplified by ULIP [78], have explored learning multimodal joint representations for 3D shapes. They train 3D-native shape encoders by aligning 3D shape embeddings with CLIP's language and/or image embeddings through multimodal contrastive learning. Works like ReCon [53] further combines cross-modal contrastive learning with masked auto-encoding for added enhancement. While these methods allow for zero-shot 3D classification through the computation of 3D-text similarity, the amount of distilled knowledge and their model capability are heavily limited by the small-scale training datasets used. Our work follows this paradigm but aims to learn more generalizable and scalable representations to enable open-world 3D shape understanding.

## 3 Method

We propose a novel method, *OpenShape*, for learning generalizable and scalable multi-modal joint representation between language, 2D images, and 3D shapes, as shown in Figure 2. We first introduce the multi-modal contrastive learning framework we used for aligning representations of three modalities in Section 3.1. We then elaborate how we create our training sets and enrich our text data in Sections 3.2 and 3.3. In Section 3.4, we present how we scale up our 3D backbone models. Finally, we propose a hard negative mining strategy to enhance contrastive learning in Section 3.5.

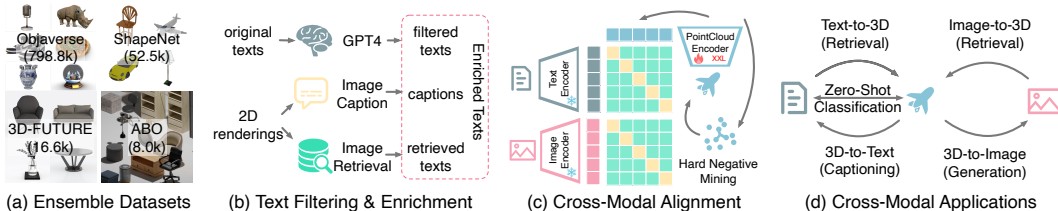

(a) Ensemble Datasets  (b) Text Filtering & Enrichment  (c) Cross-Modal Alignment  (d) Cross-Modal Applications

Figure 2: (a) We ensemble four public 3D shape datasets, resulting in 876k shapes that encompass diverse categories and concepts. (b) We propose three strategies to automatically filter and enrich the noisy texts in the original datasets. (c) We train a 3D point cloud encoder to align the 3D shape embedding space with the CLIP's text and image embedding spaces. We perform cross-modal contrastive learning with scaled 3D backbones and hard negative mining. (d) OpenShape embeddings can be easily integrated with other CLIP-based models, enabling various cross-modality tasks.

## 3.1 Multi-Modal Representation Alignment

We aim to learn 3D shape representations that are aligned with pretrained CLIP embedding spaces of language and image. As shown in Figure 2 (c), we train a 3D native encoder $f^P$ that takes a 3D point cloud as input and extracts 3D shape feature. Following previous works [53, 78, 19], such as ULIP [78], we utilize multi-modal contrastive learning for representation alignment. Since CLIP is pretrained on a much larger scale data, we freeze both its text encoder $f^T$ and its image encoder $f^I$ during feature alignment to preserve CLIP's feature priors and avoid model collapse. Specifically, given a sampled batch of triplets $\{(P_i, T_i, I_i)\}$, where $P_i$ denotes a point cloud of a 3D shape, $T_i$ and $I_i$ denote corresponding text and image, the contrastive loss is calculated as:

$$-\frac{1}{4n}\sum_i \left( \log \frac{\exp(h_i^P \cdot h_i^T/\tau)}{\sum_j \exp(h_i^P \cdot h_j^T/\tau)} + \log \frac{\exp(h_i^T \cdot h_i^P/\tau)}{\sum_j \exp(h_i^T \cdot h_j^P/\tau)} + \log \frac{\exp(h_i^P \cdot h_i^I/\tau)}{\sum_j \exp(h_i^P \cdot h_j^I/\tau)} + \log \frac{\exp(h_i^I \cdot h_i^P/\tau)}{\sum_j \exp(h_i^I \cdot h_j^P/\tau)} \right)$$
(1)

where $n$ is the number of shapes in a batch; $\tau$ is a learnable temperature; $h_i^P = f^P(P_i)/|f^P(P_i)|$, $h_i^T = g^T(f^T(T_i))/|g^T(f^T(T_i))|$, and $h_i^I = g^I(f^I(I_i))/|g^I(f^I(I_i))|$ denote normalized projected features of $P_i$, $T_i$, and $I_i$, where $g^T$ and $g^I$ are two learnable linear projections. Since $f^T$ and $f^I$ are frozen, we extract all $f^T(T_i)$ and $f^I(I_i)$ before training and cache them for acceleration. In most of our experiments, we utilize OpenCLIP ViT-bigG-14 [25] as the pretrained CLIP model.

## 3.2 Ensembling 3D Datasets

Since the scale and diversity of training triplets play a crucial role in learning scalable shape representations, we ensemble four currently-largest public 3D datasets for training as shown in Figure 2 (a), resulting in 876k training shapes. Among these four datasets, ShapeNetCore [8], 3D-FUTURE [16] and ABO [11] are three popular datasets used by prior works. They contain human-verified high-quality 3D shapes, but only cover a limited number of shapes and dozens of categories. The Objaverse [12] dataset is a more recent dataset, containing many more 3D shapes and covering significantly more diverse categories. However, shapes in Objaverse are mainly uploaded by web users and not verified by experts, and thus have uneven quality and exhibit highly unbalanced distributions, necessitating further processing.

To create triplets for training, for each shape, we sample 10,000 points from the mesh surface and interpolate the point colors according to the mesh textures. We also render 12 color images from the preset camera poses that uniformly cover the whole shape. For datasets providing thumbnails, we include them as part of image candidates, since they typically capture the shape from a better camera view. For the Objaverse dataset, we use the model name as the raw text for each shape. For other datasets, we utilize provided metadata to create raw texts (see supplementary for details). During each pretraining iteration, we randomly sample one rendered image or thumbnail for each shape, and apply standard augmentation to the point clouds [78].

## 3.3 Text Filtering and Enrichment

We find that only applying contrastive learning between 3D shapes and 2D images is insufficient to fuel zero-shot 3D classification, even when training on large-scale datasets. We conjecture that this is

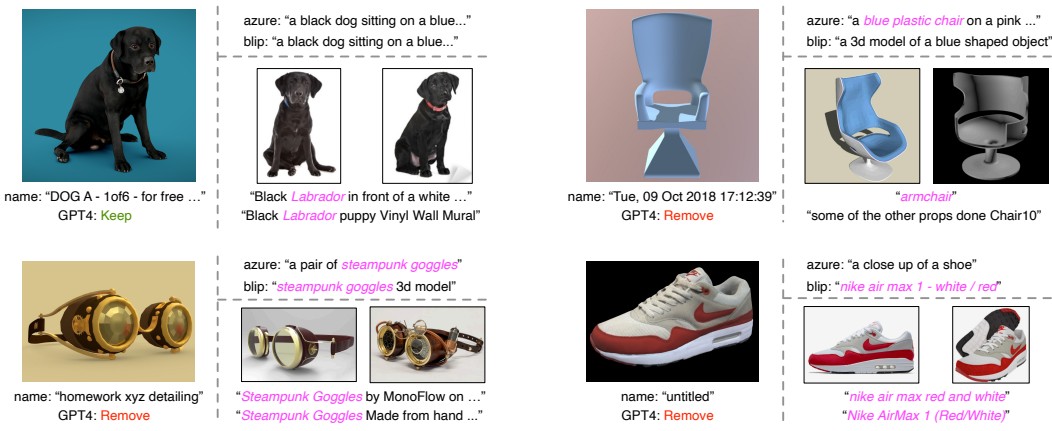

Figure 3: **Text Filtering & Enrichment Examples** In each example, the left section features the thumbnail, model name, and GPT-4 filtering results. The upper right section shows image captions from two captioning models, while the lower right section displays retrieved images and their corresponding texts.

caused by the inherent domain gap in CLIP's language and image embedding spaces, which is also observed by previous studies [37, 70]. Consequently, 3D-text alignment is not guaranteed even if we obtain good 3D-image alignments via contrastive learning. Therefore, we need to explicitly align 3D shapes with text. Along this process, to facilitate better 3D-text alignment, we introduce 3 techniques to improve the text quality: filtering, captioning, and image retrieval, as shown in Figure 2 (b).

**Filtering.** As shown in Figure 3, the 3D shapes from our main dataset, Objaverse, is dominated with noisy text descriptions ("names") uploaded by web users. Many of the problematic texts can be identified from the text itself without seeing the corresponding 3D shape. We thus leverage a powerful large language model, GPT-4 [47], to filter out inaccurate or uninformative text descriptions. We find that GPT-4 excels at recognizing irrelevant contents, such as timestamps, pure model numbers, incomprehensible descriptions, random filenames (e.g., new project), and random characters. Through GPT-4, we filter out about 30% of raw user texts. Note that we only filter the texts, and still keep all shapes for training. More details, such as the prompts we used, are presented in the supplementary.

**Captioning.** We utilize BLIP [34] and the Azure cognition services to caption the 2D thumbnails (if present, or images rendered from a fixed frontal view) of the 3D models, obtaining two texts for each shape. As shown in Figure 3, the captioning models can usually produce meaningful and descriptive captions that either enhance user-uploaded texts or replace low-quality ones. We also notice that the two caption models complement each other, leading to better performance.

**Image Retrieval.** In addition to image captioning, we also perform image retrieval to obtain additional descriptions of 3D models. We retrieve k-NN images of shape renderings from the LAION-5B dataset [65] using the CLIP ViT-L retrieval index [6]. We then take the captions of the k-NN images as the retrieved texts for our 3D models. Compared with captioning model generations, retrieved texts cover a wider range of text styles. They can also include more fine-grained semantics than both the user texts and the generated captions (e.g., "Labrador" in Figure 3).

In each iteration of pretraining, for each shape, we first randomly sample a text source category among the raw text (if unfiltered), the captions, and the retrieved texts. We then select a text candidate from the selected category. We also apply the template-based prompt engineering technique used in ULIP [78] to both training texts and test-time category names. Specifically, we extend a word or a phrase to a collection of templated simple sentences and take their average embedding.

## 3.4 Scaling Up 3D Point Cloud Backbones

Previous works on 3D point cloud learning have primarily focused on smaller-scale datasets like ShapeNet. These techniques may not be directly applicable to our larger-scale ensembled dataset and need to be scaled up accordingly. We find that different 3D backbones may exhibit distinct behavior and scalability when trained on datasets with varying sizes. Specifically, we compare six popular

Table 1: Comparison of different 3D backbones **before scaling up their parameters**. Models are trained on ShapeNet [8] or our ensembled dataset excluding Objaverse-LVIS [12]. Zero-shot classification performance are evaluated on ModelNet40 [75] and Objaverse-LVIS [12].

| Model | #Param. | Train on ShapeNet [8] | | Train on Ens-no-LVIS | |
|---|---|---|---|---|---|
| | | MNet40 | O-LVIS | MNet40 | O-LVIS |
| PointNet [51] | 1.3M | 67.0 | 9.3 | 74.9 | 24.4 |
| DGCNN [73] | 2.3M | 67.8 | 9.0 | 74.2 | 24.8 |
| PointMLP [42] | 9.3M | 73.5 | 12.9 | 82.9 | 36.6 |
| PointNeXt [54] | 2.8M | 72.6 | 12.2 | 81.6 | 33.8 |
| PointBERT [81] | 5.1M | 70.3 | 10.8 | 84.5 | 37.0 |
| SparseConv [10] | 5.3M | 70.7 | 10.6 | 78.8 | 31.7 |
| std. dev. | | 2.3 | 1.4 | 3.9 | 5.1 |

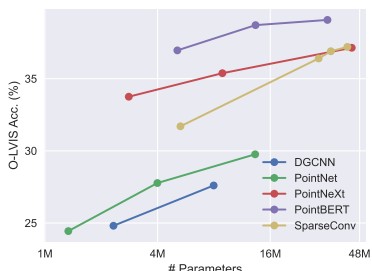

Figure 4: Accuracy on Objaverse-LVIS [12] when *scaling up* the parameters of different models.

backbones trained on ShapeNet or our ensembled dataset by evaluating their zero-shot classification performance on ModelNet40 [75] and Objaverse-LVIS datasets (for now, these backbones are trained with their original configurations and without scaling up model sizes). **Objaverse-LVIS** is a subset of Objaverse dataset with human-verified category labels. With 1,156 categories, it serves as a suitable dataset for evaluating zero-shot long-tail classification, and we exclude all shapes of Objaverse-LVIS from this experiment. Results are shown in Table 1. We find that when trained on ShapeNet, all backbones share similar performances. However, when trained on our ensembled dataset, the performance gap between backbones increases significantly. This suggests that while the original versions of these backbones share a similar number of parameters, some may have been saturated when trained on small datasets, while others do not.

We also explore the performance and scalability of these backbones when scaling up the model sizes and training on our ensembled dataset. Please refer to the supplementary for details on how we scale up each model. As shown in Figure 4, we observe that all 3D backbones benefit significantly from model scaling. However, traditional backbones without a shrinking hierarchical structure, such as DGCNN and PointNet, require operating completely on dense points or modeling the relationships (e.g., through kNN) between dense points. As a result, they become more time-consuming and memory-intensive when scaled up compared to more modern backbones. We therefore select PointBERT [81] (Transformer-based) and SparseConv [10] (convolution-based) as our 3D backbones for the remaining experiments, as they exhibit strong performance and scalability.

### 3.5 Hard Negative Mining

Our ensembled dataset exhibits a high degree of class imbalance. Certain common categories, such as building, may occupy tens of thousands of shapes, while many other categories, such as walrus and wallet, are underrepresented with only a few dozen or even fewer shapes. Consequently, when randomly constructing batches, it is unlikely that shapes from two confusing categories (e.g., apples and cherries) will be contrasted within the same batch. Inspired by some previous works [58, 30], we propose an offline hard negative mining strategy for improving the training efficiency and performance. Specifically, in the first round of training, we train our model with random batches until it is about to converge. We then compute the kNN for each shape in the learned 3D embedding space. In the second round of training, for each iteration, we randomly select $s$ seed shapes and then obtain $m$ neighbors from the kNN results of each seed shape, resulting $s \times m$ shapes per batch. In this way, confusing pairs are more likely to be selected in a single batch. However, this may also introduce false negative pairs (e.g., two apples) into contrastive learning. To mitigate this issue, we leverage image and text embeddings to filter out pairs sharing similar texts when calculating the contrastive loss. Specifically, for two shapes $i$ and $j$ selected from the same seed shape, if $h_j^T \cdot h_i^I + \delta > h_i^T \cdot h_i^I$, where $h^T$ and $h^I$ are text and image embeddings, and $\delta$ is a small threshold, we believe that the text embeddings of $i$ and $j$ are very close to each other, and we remove $j$ from $i$'s negative examples when calculating contrastive loss. By employing this strategy to construct batches, we observe faster and better model learning.

Table 2: Zero-shot classification on Objaverse-LVIS [12], ModelNet40 [75], and ScanObjectNN [70].

| Method | training shape source | Objaverse-LVIS [12] | | | ModelNet40 [75] | | | ScanObjectNN [71] | | |
|---|---|---|---|---|---|---|---|---|---|---|
| | | Top1 | Top3 | Top5 | Top1 | Top3 | Top5 | Top1 | Top3 | Top5 |
| PointCLIP [86] | 2D inferences, | 1.9 | 4.1 | 5.8 | 19.3 | 28.6 | 34.8 | 10.5 | 20.8 | 30.6 |
| PointCLIP v2 [88] | no 3D training | 4.7 | 9.5 | 12.9 | 63.6 | 77.9 | 85.0 | 42.2 | 63.3 | 74.5 |
| ReCon [53] | | 1.1 | 2.7 | 3.7 | 61.2 | 73.9 | 78.1 | 42.3 | 62.5 | 75.6 |
| CG3D [19] | | 5.0 | 9.5 | 11.6 | 48.7 | 60.7 | 66.5 | 42.5 | 57.3 | 60.8 |
| CLIP2Point [24] | | 2.7 | 5.8 | 7.9 | 49.5 | 71.3 | 81.2 | 25.5 | 44.6 | 59.4 |
| ULIP-PointBERT (Official) [78] | ShapeNet | 6.2 | 13.6 | 17.9 | 60.4 | 79.0 | 84.4 | 51.5 | 71.1 | 80.2 |
| OpenShape-SparseConv | | 11.6 | 21.8 | 27.1 | 72.9 | 87.2 | 93.0 | 52.7 | 72.7 | 83.6 |
| OpenShape-PointBERT | | 10.8 | 20.2 | 25.0 | 70.3 | 86.9 | 91.3 | 51.3 | 69.4 | 78.4 |
| ULIP-PointBERT (Retrained) | | 21.4 | 38.1 | 46.0 | 71.4 | 84.4 | 89.2 | 46.0 | 66.1 | 76.4 |
| OpenShape-SparseConv | Ensembled (no LVIS) | 37.0 | 58.4 | 66.9 | 82.6 | 95.0 | 97.5 | 54.9 | 76.8 | 87.0 |
| OpenShape-PointBERT | | 39.1 | 60.8 | 68.9 | **85.3** | 96.2 | 97.4 | 47.2 | 72.4 | 84.7 |
| ULIP-PointBERT (Retrained) | | 26.8 | 44.8 | 52.6 | 75.1 | 88.1 | 93.2 | 51.6 | 72.5 | 82.3 |
| OpenShape-SparseConv | Ensembled | 43.4 | 64.8 | 72.4 | 83.4 | 95.6 | 97.8 | **56.7** | 78.9 | 88.6 |
| OpenShape-PointBERT | | **46.8** | **69.1** | **77.0** | 84.4 | **96.5** | **98.0** | 52.2 | **79.7** | **88.7** |

# 4 Experiments

## 4.1 Zero-Shot Shape Classification

We evaluate the zero-shot classification performances of our models on three benchmarks: the traditional ModelNet40 [75] and ScanObjectNN [71], as well as a new benchmark, Objaverse-LVIS [12]. ModelNet40 and ScanObjacetNN consist of 40 and 15 common categories, respectively. Objaverse-LVIS is an annotated subset of Objaverse [12] and comprises 46,832 shapes among 1,156 LVIS [17] categories. With a much larger base of classes than other benchmarks, Objaverse-LVIS presents a challenging long-tailed distribution, making it a better reflection on models' performance in open-world scenarios. We compare OpenShape with existing zero-shot approaches, including PointCLIP [86], PointCLIPv2 [88], ReCon [53], CG3D [19], CLIP2Point [24], and ULIP [78]. Among them, PointCLIP [86] and PointCLIPv2 [88] project point clouds into 2D images and directly utilize 2D CLIP for inference, while other methods leverage the CLIP embedding spaces for alignment and require 3D shapes for training. We report results on these baselines using their released checkpoints. To better analyze the source of our performance gains, we also retrain the baseline ULIP [78] on our ensembled shape dataset, but we use the original texts in the four constituent datasets along with the official codebase without backbone scaling. We train OpenShape and ULIP on three different sets of training shapes: "**Ensembled**" denotes using all shapes from the four datasets; "**Ensembled (no LVIS)**" is the same but excludes all shapes from the Objavserse-LVIS subset; "**ShapeNet**" only includes shapes from the ShapeNet [8] dataset. Note that even when LVIS shapes are included in the training shapes (i.e., the "Ensembled" dataset), their test-time category labels are probably not included in their training texts. Please refer to the supplementary for more training and evaluation details.

Table 2 shows the results. We observe that OpenShape consistently outperforms prior approaches, even when trained only on ShapeNet. When models are trained on our larger-scale ensembled dataset, they receive a significant performance boost. In this case, OpenShape still surpasses retrained ULIP by a significant margin, demonstrating the advantages of our text enrichment, backbone scaling, and other training strategies. Specifically, OpenShape greatly improves the classification accuracy on the long tail categories in Objaverse-LVIS from a dull $< 10\%$ to $46.8\%$, outperforming the retrained ULIP by about 20 points and reaching a decent top-5 accuracy of $77.0\%$. These results demonstrate OpenShape's capability to recognize open-world objects effectively. As for ModelNet40, OpenShape achieves a $85.3\%$ accuracy, surpassing previous methods by a substantial margin of at least 20 percent. OpenShape also achieves impressive top-3 and top-5 accuracies of $96.5\%$ and $98.0\%$. To the best of our knowledge, this is the first time zero-shot methods have matched the performance of a fully-supervised 3D learning method on ModelNet40, where OpenShape outperforms fully-supervised 3D ShapeNets [75] and VoxNet [43]. In addition, on ScanObjectNN, which contains challenging real scans with noise and occlusion, OpenShape exhibits decent sim-to-real transfer capabilities. To contextualize, OpenShape-SparseConv achieves $56.7\%$ zero-shot accuracy on ScanObjectNN without specific sim-to-real training, which surpasses $52.7\%$ reported by SKPConv [74], a recent method specially designed for sim-to-real transfer in point cloud classification tasks.

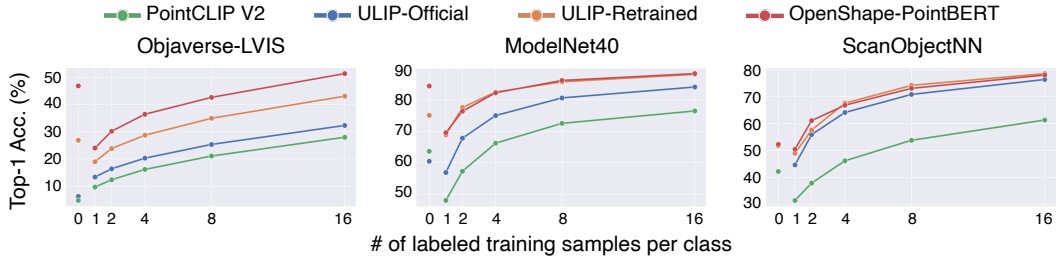

Figure 5: Few-shot linear probing on Objaverse-LVIS [12], ModelNet40 [75], and ScanObjectNN [70]. We report the average performance over 10 random seeds.

Table 3: Ablation study. Top 1 zero-shot accuracies on ModelNet40 [75] and Objaverse-LVIS [12] are shown.

| Variant | O-LVIS | MNet40 |
|---|---|---|
| No Objaverse shapes | 13.9 | 75.5 |
| Only Objaverse shapes | 41.6 | 79.2 |
| No backbone scale up | 31.7 | 78.7 |
| No caption & retrieval | 37.0 | 82.9 |
| No text filtering | 41.4 | 82.9 |
| No point rgb, only xyz | 39.6 | 83.6 |
| No text contras. learning | 23.3 | 67.4 |
| No image contras. learning | 41.0 | 81.0 |
| Full | 42.0 | 83.1 |
| Full + hard mining | 43.4 | 83.4 |

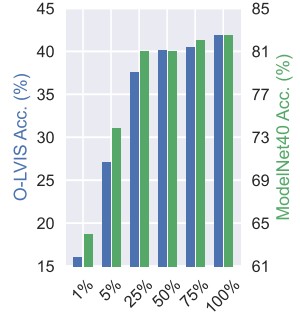

Figure 6: Ablation study on using different ratios of training data.

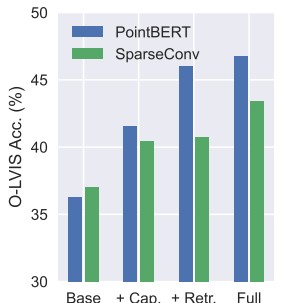

Figure 7: Ablation study on different text enrichment strategies.

## 4.2 Few-Shot Linear Probing

In the literature, linear probing is a common way to assess the representation learning capabilities of a model. To perform linear probing, we gather and freeze the representation vectors from all samples in a dataset. Subsequently, we train a linear classifier using these fixed vectors and few-shot class labels. We evaluate the accuracy of the linear classifier on three benchmarks: Objaverse-LVIS [12], ModelNet40 [75], and ScanObjectNN [71]. Figure 5 summarizes the performance of OpenShape in comparison with ULIP [78] (official release and our retrained versions) and PointCLIPv2 [88]. On the most challenging Objaverse-LVIS benchmark, OpenShape outperforms all other methods by a large margin. Notably, zero-shot OpenShape beats few-shot linear probes of other methods. On ModelNet40 and ScanObjectNN, we do not see a large performance margin between OpenShape and retrained ULIP. We hypothesize that for few-shot ModelNet40, the error is dominated by in-category sample bias rather than the representation quality; while for ScanObjectNN, the domain gap plays a major role. Since both OpenShape and retrained ULIP are exposed to the same source domain of training objects, their few-shot out-of-domain generalization performances tend to be similar.

## 4.3 Ablation Study

We perform various ablations by training a scaled version of SparseConv [10] on the ensembled dataset and then evaluate it on the Objaverse-LVIS [12] and ModelNet40 [75] zero-shot classification benchmarks, unless otherwise specified. The results are shown in Table 3 and Figures 6 and 7.

**Data and Model Scaling.** We investigate the impact of training data by ablating (1) without or with only Objaverse shapes (Tab. 3) and (2) with different ratios of our ensembled dataset (Fig. 6). We observe that training with 1% of our ensembled dataset (about 8.8k shapes) achieves similar or better zero-shot performance than training without Objaverse shapes (about 77.1k shapes), indicating that the diversity of training data is sometimes more crucial than the scale. In addition, we compare the performances between scaled-up and non-scaled-up backbones. From Tab. 3, we demonstrate that model scaling plays an essential role when training on our large-scale ensembled dataset (also Fig. 4).

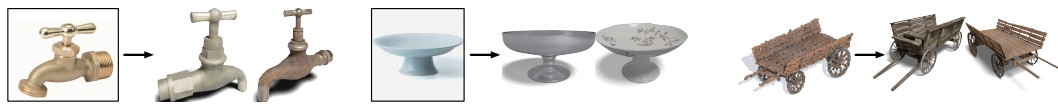

Figure 8: 3D shape retrieval from image (left, mid) and point cloud (right).

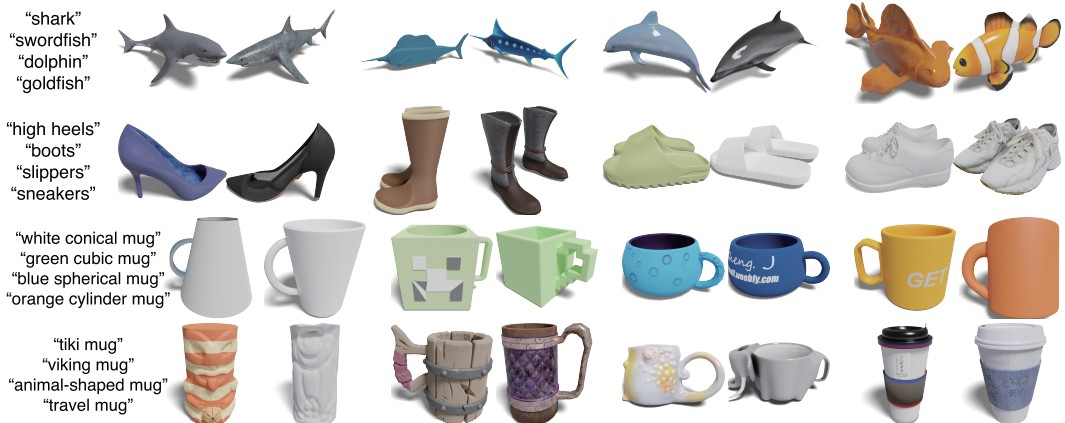

Figure 9: **Text-input 3D shape retrieval.** In each row, we show input texts on the left and two retrieved shapes for each text on the right. OpenShape embedding encodes a wide range of visual and semantic concepts and enables (a) retrieval of fine-grained subcategories (first two rows), and (b) control of attributes (e.g., color, shape, style) and their combinations (last two rows).

**Text Filtering and Enrichment.** As shown in Tab. 3, both text filtering and text enrichment are beneficial for performance. We also investigate the specific text enrichment strategies to use for the SparseConv and PointBERT backbones. In Fig. 7, we observe that both image captioning and text retrieval are helpful, and including both yield the best results. Notably, PointBERT improves more than 10 points from text enrichment, highlighting the significance of enhancing text quality.

**Other Aspects.** We also conduct additional ablation studies on color information, contrastive loss components, and our hard-negative mining strategy in Tab. 3. We observe that OpenShape performs well with only $xyz$ coordinates as input and no RGB color. While 3D-image contrastive loss is also helpful, we observe that 3D shape-text alignment plays a very essential role for model zero-shot generalization, which necessitates our text filtering and text enrichment strategies that significantly enhance text quality. Lastly, by employing our hard negative mining strategy, OpenShape effectively addresses the issue of unbalanced data distribution, leading to further improvements in performance.

## 4.4 Cross-Modal Applications

**Multi-modal 3D Shape Retrieval.** Through OpenShape multi-modal representations, we can index and retrieve 3D shapes from images, texts, or point clouds. In this section, we retrieve 3D shapes from our ensembled dataset by calculating the cosine similarity between input embedding(s) and 3D shape embeddings and performing kNN. As shown in Figure 8, OpenShape is capable of retrieving visually or semantically similar shapes from a single image or point cloud input. OpenShape embeddings encode a wide range of visual and semantic concepts. In Figure 9, we show that OpenShape supports retrieving 3D shapes from detailed text descriptions, which include fine-grained subcategories, attributes, and their combinations. Note that these input texts are typically not present in the raw texts of the retrieved shapes, indicating that OpenShape effectively learns generalizable concepts across shapes. In Figure 1, we provide a demo which takes two 3D shapes as inputs and retrieves the shapes that are simultaneously closest to both inputs. This is achieved by finding $\arg\max_i \min(h_i^P \cdot h_a^P, h_i^P \cdot h_b^P)$, where $h_a^P$ and $h_b^P$ denote normalized shape embeddings of the two input shapes. We can see that the retrieved shapes integrate visual or semantic elements in an interesting manner, highlighting the rich concepts and priors encoded in OpenShape embeddings.

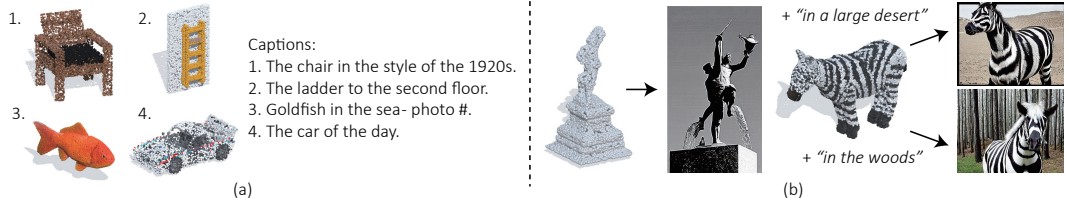

Figure 10: **(a) Point cloud captioning. (b) Point cloud-conditioned image generation.** Our learned 3D shape embeddings can be integrated with off-the-shelf pretrained CLIP-based models (e.g., captioning and image generation models) to support various cross-modal applications.

**Shape-Conditioned Multimodal Generation.** As OpenShape's 3D shape representations are aligned with CLIP's image and text embedding spaces, they can serve as inputs into other CLIP-based models to facilitate various multimodal generation applications. For example, we show that by feeding our 3D shape embeddings into ClipCap [46], an off-the-shelf image captioning model, along with Stable unCLIP [56], a text-to-image diffusion model, we can perform point cloud captioning and point cloud-conditioned image generation (optional text prompt supported) without extra training or finetuning. Qualitative results are shown in Figure 10. Please refer to the supplementary for more results and details.

## 5    Limitation and Conclusion

We introduce OpenShape, a novel approach for learning scalable and generalizable multi-modal joint representations for 3D shapes. OpenShape representations effectively capture a wide range of semantic and visual concepts, enabling superior capabilities for open-world 3D shape recognition. By aligning OpenShape with CLIP's embedding space, our shape embeddings can be integrated with off-the-shelf CLIP-based models for various cross-modality applications. Moving forward, there are several directions worth further exploration: (a) More 3D data. While we utilized 876k 3D shapes during training, this is still quite limited compared to the 2D counterparts. We hope that our work inspires future investments in more resources to build even more powerful 3D representations. (b) Part-level information. Our current shape representations mainly focus on global semantic and visual features, and it would be beneficial to add more part-level supervision during training. (c) Sim-to-real domain gap. Our model is mainly trained on synthetic data, and it's challenging but crucial to explore explicit designs for reducing the domain gap with real-world shapes.

## Acknowledgments

This work is supported in part by gifts from Qualcomm.

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

# A More Examples of Multi-Modal 3D Shape Retrieval

In Figures 11 and 12, we showcase more examples of multi-modal 3D shape retrieval.

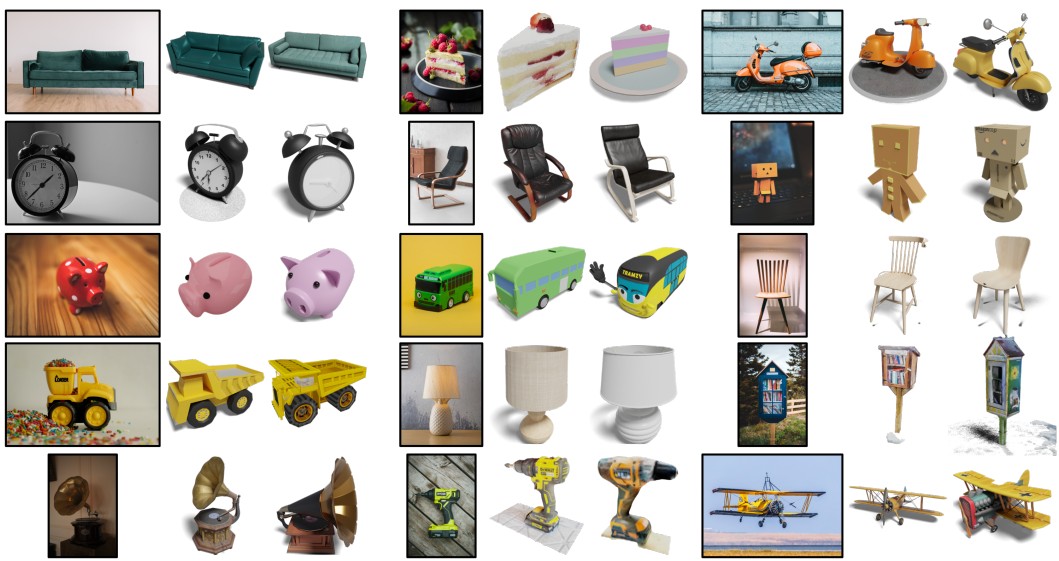

Figure 11: **Image-input 3D shape retrieval.** In each triplet, we present the input image and two 3D shapes retrieved using OpenShape embeddings from the Objaverse [12] dataset. Input images are from `unsplash.com`.

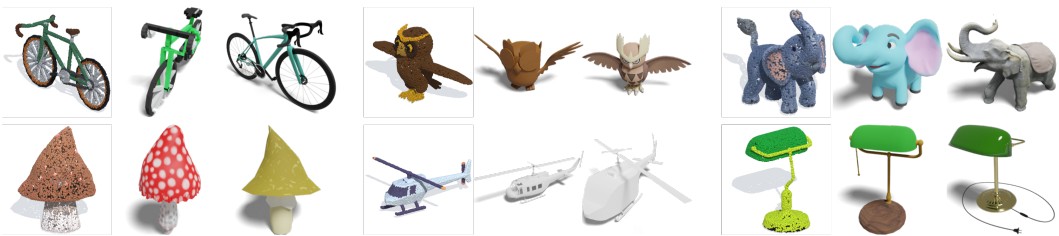

Figure 12: **Point cloud-input 3D shape retrieval.** In each triplet, we present the input point cloud and two 3D shapes retrieved using OpenShape embeddings from the Objaverse [12] dataset.

# B More Examples of Shape-Conditioned Multimodal Generation

In Figure 13 and Figure 14, we showcase more examples of point cloud captioning and point cloud-conditioned image generation.

# C Details on Raw Text Generation and Filtering

## C.1 Raw Text Generation

We leverage the metadata from the four datasets to generate the raw texts. Although the original datasets may contain numerous attributes for each shape, we carefully choose the most informative ones to compose the text, ensuring its quality and relevance.

**Objaverse**: We utilize the `name` associated with each shape to serve as the text.

**ShapeNetCore**: For each shape, we generate three types of texts: (a) the `name`, (b) the `category name` (with a total of 55 categories), and (c) the concatenation of the `sub-category names` (with a total of 336 sub-categories), separated by commas.

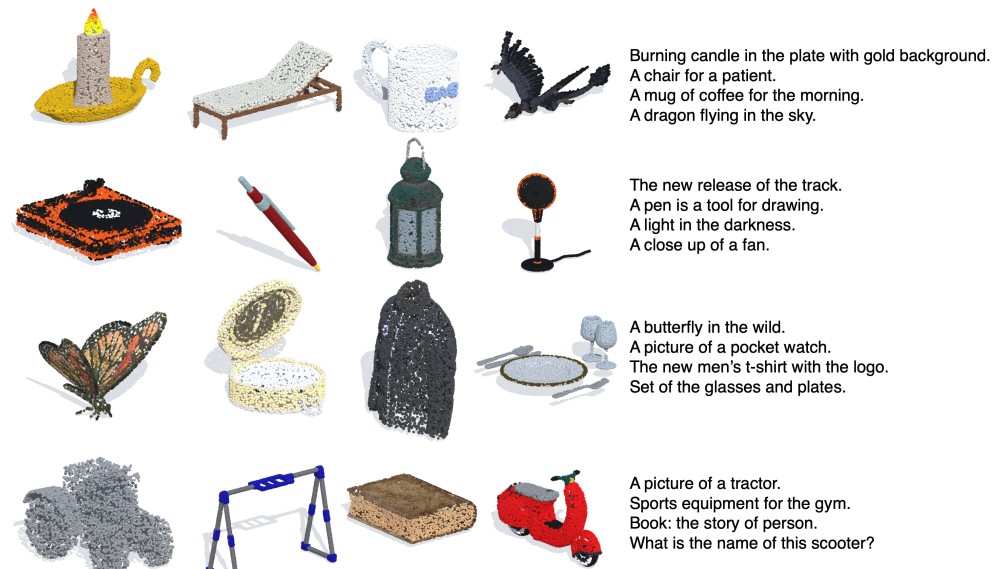

Burning candle in the plate with gold background.
A chair for a patient.
A mug of coffee for the morning.
A dragon flying in the sky.

The new release of the track.
A pen is a tool for drawing.
A light in the darkness.
A close up of a fan.

A butterfly in the wild.
A picture of a pocket watch.
The new men's t-shirt with the logo.
Set of the glasses and plates.

A picture of a tractor.
Sports equipment for the gym.
Book: the story of person.
What is the name of this scooter?

Figure 13: **Point cloud captioning**. In each row, we show the input point clouds on the left and the generated captions on the right.

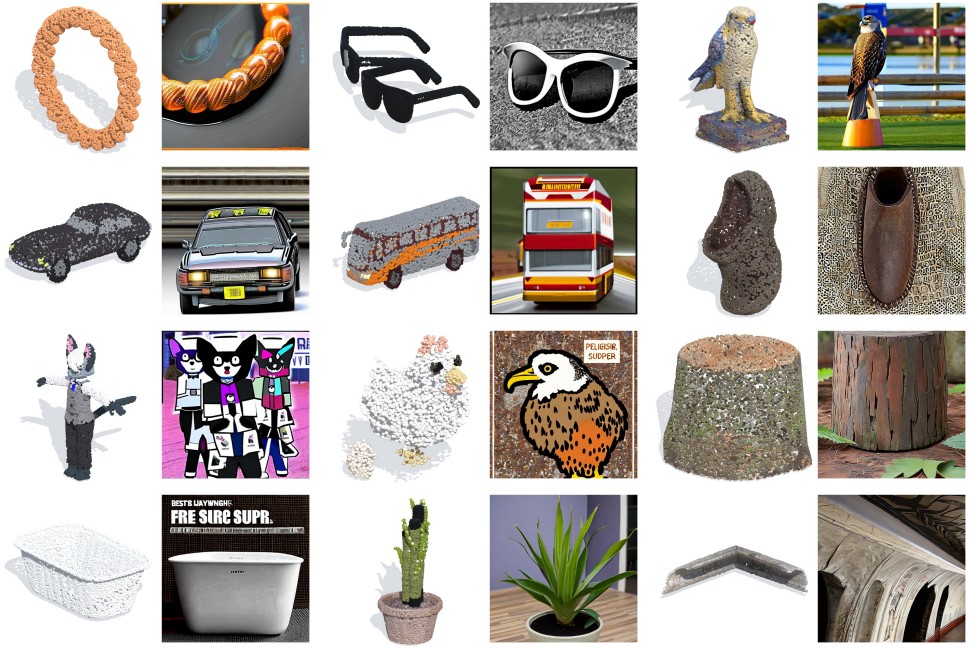

Figure 14: **Point cloud-conditioned image generation**. Each row shows three examples (input point clouds and generated images).

**3DFuture**: For each shape, we generate two types of texts: (a) the `category`, and (b) the concatenation of `category`, `style`, `theme`, and `material`, separated by commas.

**ABO**: For each shape, we generate two types of texts: (a) the `item_name`, and (b) the `product_type`.

In this way, we generate one or more raw texts for each shape.

## C.2 Raw Text Filtering

> I am analyzing a 3D dataset with various text descriptions for the 3D models. However, many of these texts are inaccurate or uninformative, and therefore, not suitable as descriptions for 3D models. I need your help to identify such incorrect texts. Specifically, if a text primarily consists of irrelevant or uninformative content, such as timestamps, model numbers, incomprehensible descriptions, random filenames (e.g., "my project"), random characters, etc., please respond with "N". If a text contains a clear noun (or noun phrase) that could potentially describe a 3D object, please respond with "Y". You will find a list of texts below, and each line contains a three-digit ID and associated text. For each text, please respond with "Y" or "N", following the ID number (e.g., "001 Y" or "002 N"). Please evaluate all 256 texts.
>
> 000 New project ( 19 )
> 001 3December - Chemestry
> 002 Fake Brand Soda Can
> 003 Spartan Shild
> 004 Apple3d
> 005 Landmine
> 006 FaunveinB-S
> 007 FIGURA 5
> 008 Sphero Blue
> 009 Sofa
> 010 Maddox
> 011 A3 Complete
> 012 Suspension Bridge
> 013 Maung
> 014 Captain-americas-shield
> 015 sphorb4
> ......

| |
| :--- |
| 000 N |
| 001 Y |
| 002 Y |
| 003 Y |
| 004 Y |
| 005 Y |
| 006 N |
| 007 N |
| 008 Y |
| 009 Y |
| 010 N |
| 011 N |
| 012 Y |
| 013 N |
| 014 Y |
| 015 N |
| ...... |

We employ GPT-4 [47] to filter out uninformative raw texts. To accomplish this, we divide all the raw texts into batches, each containing 256 entries, and process each batch independently using GPT-4. Here is an example illustrating the prompt we used and the corresponding response generated by GPT-4.

Afterwards, we combine all the responses to create the final filtering results, effectively removing approximately 30% of the raw texts.

## D   Details on the Backbone Scaling Experiment

In Figure 4 of the main paper, we investigate the performance and scalability of various backbones when scaling up their model sizes. For this experiment, we employ a default resolution of 10,000 points for input point clouds, a batch size of 200, and conduct the experiment on a single A100 GPU. In general, if instructions are given in the original paper of a backbone, we scale up the model as instructed. Otherwise, we scale up the model by expanding width or depth (i.e., stacking blocks or layers). Specifically, we scale up each backbone as follow:

**PointBERT [81]**   The scaling parameters are shown in Table 4. We scaled PointBERT to 72.1M parameters beyond the 32.3M version reported in Figure 4 of the main paper. However, at this scale, the model dramatically overfits on the training data and performs worse on all benchmarks than the 32.3M version.

**SparseConv [10]**   The smallest version (5.3M parameters) of the model is adapted from the MinkowskiFCNN model by adjusting the width of the final convolution and linear layers. The remaining three models are adaptations of MinkowskiResNet, each varying in the number of basic ResNet blocks used. See Table 5 for the specific scaling parameters.

Table 4: Hyperparameters for scaling up PointBERT [81].

| # Parameters | # Layers | Width | # Heads | MLP Dim | # Patches | Patch Embed Dim |
|---|---|---|---|---|---|---|
| 5.1M | 6 | 256 | 4 | 1024 | 64 | 96 |
| 13.3M | 6 | 512 | 8 | 1024 | 64 | 128 |
| 32.3M | 12 | 512 | 8 | 1536 | 384 | 256 |
| 72.1M | 12 | 768 | 12 | 2304 | 512 | 256 |

Table 5: Hyperparameters for scaling up SparseConv [10].

| # Parameters | # Convolution Layers | # Linear Layers |
|---|---|---|
| 5.3M | 7 | 4 |
| 29.0M | 18 | 3 |
| 33.7M | 26 | 3 |
| 41.3M | 42 | 3 |

**PointNeXt [54]**  PointNeXt is proposed as a scalable version of PointNet++ [52], and includes S/B/L/XL variants in the original paper. We simply adopt these official configurations.

**DGCNN [73] and PointNet [51]**  For these two backbones without a hierarchical structure, we increase the width of each layer proportionally to scale up to 4xPointNet and 2xDGCNN before we hit the GPU memory limit. As the models operate completely on dense points, it is impractical to use the default 10k-point resolution. We thus reduce the input resolution for the two backbones, resulting in 1k points for DGCNN and 4k points for PointNet.

## E  Details on Training and Evaluation

**Training Details**  We freeze the CLIP text and image encoders and train the 3D encoder and two projection heads on our ensembled dataset using the cross-modal contrastive loss. We train the model on a single A100 GPU with a batch size of 200. Since we precache the text and image CLIP embeddings of all shapes, the training is greatly accelerated and takes about 300 A100 hours for convergence. We utilize an exponential learning rate schedule, and employ an range test to find the initial learning rate. For 32.3M version of PointBERT, we utilize a learning rate of $5e - 4$; for 72.1M version of PointBERT, we utilize a learning rate of $4e - 4$; and for other models, we utilize a learning rate of $1e - 3$. For hard-negative mining, the number of seed shapes $s$ is set to 40, and the number of neighbors $m$ is set to 5 per shape, and the threshold $\delta$ is set to 0.1.

**Fine-tuning CLIP Text and Image Encoders?**  After training OpenShape-PointBERT, we conducted experiments to unfreeze and finetune the CLIP text encoder for a single epoch. However, the results obtained did not demonstrate any noticeable improvement on the benchmarks. Moreover, we observed that finetuning the CLIP text encoder could potentially undermine the generalization capabilities of CLIP and hinder the integration of OpenShape embeddings into existing CLIP-based models. As a result, we choose to freeze the CLIP encoders throughout the entire training process.

**Evaluation Details**  We evaluated all baselines using their publicly released pretrained checkpoints. Additionally, we retrained ULIP [78] on our ensembled training shapes using their official code base and backbone networks. Note that the retrained ULIP model utilized the original raw texts from the four datasets during training (prompt engineering is also applied), rather than our filtered and enriched texts. For ModelNet40 [75], the evaluation is conducted on the test split with 2,468 shapes. Regarding ScanObjectNN [71], we follow ULIP [78] to evaluate on the OBJ_ONLY version, which contains 581 test shapes. For Objaverse-LVIS [12], the input is 10,000 sampled points with point colors. For ModelNet40 [75], the input is 10,000 sampled points without color. For ScanObjectNN [71], we utilize the official 2,048 points without color as input. All methods use the same input during evaluation. The forward inference time on an A100 GPU for a 10,000-point point cloud is approximately 0.9ms for OpenShape-SparseConv and 3.8ms for OpenShape-PointBERT.

# F Details on Shape-Conditioned Multimodal Generation

**Point Cloud Captioning** CLIPCap [46] utilizes a 10-token prefix generated from CLIP image embeddings to enable GPT-2 for captioning. In order to align with the off-the-shelf CLIPCap model, we trained a variant of OpenShape-PointBERT that employs CLIP ViT-B/32 embeddings instead of OpenCLIP ViT-G/14 used in other experiments. Consequently, we directly input the point cloud encoding, *without normalization*, into CLIPCap for captioning.

**Point Cloud Conditioned Image Generation** We take the Stable Diffusion v2.1 unCLIP model [56] for image generation and replace the CLIP image condition encoder with our OpenShape encoder to perform image generation conditioned on point clouds (and optionally text prompts). The unCLIP model takes CLIP ViT-L/14 embeddings without normalization as input. To match the embedding space, we trained a variant of OpenShape-PointBERT with CLIP ViT-L/14 embeddings. Additionally, we noticed a significant mismatching of scales ($L_2$-norm of embedding vectors) between ViT-L/14 image embeddings and OpenShape embeddings. To mitigate this issue, we perform a re-normalization on OpenShape embeddings to a $L_2$-norm of $\frac{1}{2}\sqrt{768}$, which is our observed mean $L_2$-norm of ViT-L/14 image embeddings. We use 50 diffusion steps. The guidance scale can be tuned freely.

