# OpenReview forum: "OpenShape: Scaling Up 3D Shape Representation Towards Open-World Understanding"
_NeurIPS.cc/2023/Conference — NeurIPS 2023 poster_

### Official Review · Reviewer_zuqY · 2023-06-25

**Soundness:** 4 excellent
**Presentation:** 3 good
**Contribution:** 3 good
**Rating:** 7
**Confidence:** 4

**Summary:**

The paper proposes a new system for learning a joint embedding space for text, images, and 3D shapes (point clouds). It starts from pre-trained CLIP embeddings for text and images, and learns one more embedding function for 3D shapes in a self-supervised manner, using a contrastive loss. Several additional improvements are proposed, including combining several datasets, cleaning some of those including via automated re-labelling of Objavrese, and optimisation of the 3D embedding neural network architecture. Because of these improvements, the new embedding performs significantly better than prior works on strand tasks such as "zero-shot" classification.



**Strengths:**

* The paper contributes sound incremental work on learning multi-modal embedding functions for text, images and 3D shapes. Various engineering dimensions are considered (model scaling, data diversity, scaling and quality, etc) and some insights are provided on how each of these can be improved, at least compared to current baselines. I expect these findings to provide useful guidelines for future work in this area.

* The paper does contain some interesting suggestions on how to improve datasets such as Objaverse to train multi-modal embeddings. The scheme where captions are assessed via GPT-4 and then replaced using BLIP if needed is useful.



**Weaknesses:**

* The paper is mostly about good engineering, but there isn't a lot of very deep technical innovations, or qualitatively surprising findings. The main finding, in fact, is that this is an area where scaling is still very limited (mostly due to the lack of suitable training data), and that, thus, scaling is where most of the low-hanging fruits can be had. This is a good but not very surprising message.

* The model is trained on synthetic datasets and looses much of its edge when applied to real data, compared to prior models like ULIP that are otherwise suboptimal when tested on synthetic data (e.g., Figure 5, right panel).

* Likewise, there isn't a very clear dominance of the two proposed backbones, SparseConv and PontBert: their performance swings significantly depending on the testing data in Table 2.



**Questions:**

* Filtering bad positive samples in contrastive the loss using the existing text and image embeddings (line 213) is simple but also likely useful, however it is not clear to me if this idea is ablated in Table 3. Can you clarify?

* Why SparseConv and PointBERT are so different in performance depending on the test dataset? Is there a recognisable pattern?



**Limitations:**

The paper does address some limitations, particularly in the discussion and conclusions where they are implicitly recast as "future work".

There is no discussion of ethics, although this does not seem to be an issue for this paper (notoriously, though, dataset like ShapeNet had issues with copyright, and I don't believe ObjaVerse is necessarily immune from such controversies either).

---

> ### Author Rebuttal · Authors · 2023-08-10
>
> Thank you for your insightful comments and valuable suggestions. We will revise our paper based on your feedback. Here are our responses to your comments:
>
> **Limited qualitatively surprising findings**
>
> We kindly refer the reviewer to the supplementary for more exciting results, where we showcase a lot of cross-modal applications enabled by our powerful representations, including (a) image and point cloud input 3D shape retrieval (Figures 1, 8, S1, S2), (b) text input 3D shape retrieval (Figure 9), which supports fine-grained subcategories, attributes, and their combinations; (c) point cloud captioning (Figure 10 and S3); (d) point cloud conditioned image generation (Figure 10 and S4). These applications demonstrate a broad range of semantic and visual concepts encoded in our representations and the capability of open-world 3D shape understanding, which has not been shown in previous works.
>
> With more and more large-scale 3D datasets becoming available (e.g., Objaverse-XL has 10M shapes), we believe that learning a powerful 3D representation and addressing the challenges that arose during the scaling up (the goal of this paper) may become more important and receive more attention. Open-world 3D shape understanding is more likely to be realized than ever before.
>
> **Lose much of its performance edge when applied to scanned data**
>
> We agree that the performance gap between our method and some baseline methods is small on ScanObjectNN. This is because the performance of ScanObjectNN is mainly dominated by the domain gap (e.g., noise from scanned objects). Since both models don’t have explicit designs for reducing the domain gap, they share similar performance. As mentioned in the “Conclusion and Discussion”, exploring explicit designs (e.g., data augmentation) to reduce the domain gap with scanned objects is challenging but critical, and we leave it as future work.
>
> **The performance of PointBERT and SparseConv**
>
> We did not expect a significant performance advantage between the two backbones. Overall, PointBERT consistently has a slight advantage on clean data (Objaverse-LVIS and ModelNet40), as shown in Table 2 of the main paper. Their ranking changes only when it comes to some ScanObjectNN evaluation settings. We conjecture that the convolution-based network architecture (e.g., SparseConv) is more robust than transformer-based network architecture (e.g., PointBERT) and has better generalization ability to the noise and incompleteness of real scanned objects.
>
> **Filtering bad negative pairs**
>
> The idea of using existing text and image embeddings to filter false negative pairs is important in hard negative mining since we utilize KNN shapes to build batches, and it’s highly likely to encounter false pairs in a batch (e.g., two apples in a batch). Without the filtering, the performance of hard negative mining will drop significantly in our experiments. However, the filtering is not necessary in our first-stage normal training, where batches are randomly constructed. Our experiments show that the filtering doesn’t affect the performance a lot for random batches since it’s less likely to encounter two apples in a batch.

---

> > ### Comment · Reviewer_zuqY · 2023-08-11
> >
> > I thank the reviewers for their response. I was voting for accepting the paper before, and I remain in favour.

---

### Official Review · Reviewer_quLe · 2023-06-28

**Soundness:** 3 good
**Presentation:** 3 good
**Contribution:** 3 good
**Rating:** 6
**Confidence:** 4

**Summary:**

The paper studies the problem of multi-modal learning of text, image and shapes. Shapes are represented by pointclouds and the learning is driven by the standard contrastive loss. The main technical difference of this work to prior ones lies in the scale of data and corresponding training strategies. The authors train their model on a much larger scale of data which pools Objaverse, ShapeNet, 3D-Future and ABO. To handle the noise in the text (mainly from the less-curated Objaverse), the authors use several large pretrained models to clean up the text and make it better aligned with the shapes. Hard-mining is also leveraged to make the model train better under class imbalance. With these techniques which make the model scales better with data size, the proposed model achieves the new SOTA performance across several relevant tasks and datasets.

**Strengths:**

1. The model in this paper achieves significantly better results than previous SOTA in zero-shot shape classification. It also demonstrates great qualitative performance in shape retrieval, shape interpolation, shape captioning and has good potential in other cross-modal applications as well. Such model will be practically very useful.
2. The benchmark and the findings in the paper about different models' scaling performance are valuable information to the community and future research along this direction.
3. The paper is well-written and easy to follow.

**Weaknesses:**

1. The evaluation of the zero-shot shape classification on Objaverse-LVIS might not actually be zero-shot. Based on the description in L235, even the "Ensembled (no LVIS)" can include LVIS categories, because seemingly the author only excludes the exact evaluation samples from the training set. Ideally, all shapes from the evaluation *categories* should be removed from the training set. This makes the real zero-shot generalization performance of the model potentially lower than reported and the overall comparison on this test set less informative. Besides, ModelNet40 and ScanObjectNN can also have overlapping categories with the training set. (If the authors think category overlapping does not violate their definition of zero-shot, it should be clearly stated so / "zero-shot" should be clearly defined at the beginning.)

2. The linear probe results show the proposed model performs similarly to ULIP-retrained on ModelNet and ScanObjectNN. The authors try to explain it with in-category sample bias (what does this mean?) and domain gap dominance, without providing any evidence for these hypotheses. My concern here is that, given that OpenShape performs much better than ULIP in shape classification (requires both shape and language rep) but not linear probe (only requires shape rep), the actual underlying reason could be that the shape representation is of similar quality when tested on out-of-domain, and the real difference between these models lies in the text representation. I believe it would be quite beneficial to provide more analyses on this result, as this is central to the "shape representation learning" story of the paper. For example, such analyses could be visualizing and comparing the latent structure of the shape representation; or simply performing shape-latent based NN query and evaluate the distance between queried shapes; or measuring the shape latent's alignment with image latent, to name a few.

3. The limitation of the model is not discussed at all. This is a very important aspect for readers to thoroughly understand the contribution of this paper.

**Questions:**

Please see the weaknesses for my main questions and concerns. Overall I like the paper as it presents useful findings for the community and the model's performance is strong. But I think more analyses and discussions are necessary to solidify some experiments and the overall claim.

Another relatively minor question about the result:
In Table 2, the retrained ULIP achieves 26.8% top 1 while OpenShape achieves 43.4%. The authors attribute this to "text enrichment, backbone scaling, and other training strategies" (L243). However, it is shown in the ablation that removing text enrichment leads to 6.4% drop and hard mining leads to 1.4% drop. Does this mean the main improvement over ULIP comes from the larger number of parameters/better backbones as the overall drop is ~17%?

**Limitations:**

The limitations and failure cases are not discussed and should be included.

---

> ### Author Rebuttal · Authors · 2023-08-10
>
> Thank you for your insightful comments and valuable suggestions. We will revise our paper based on your feedback. Here are our responses to your comments:
>
> **Definition of "zero-shot"**
>
> Thanks for pointing this out. We agree that the current "zero-shot" evaluation does not exclude all shapes in the evaluation categories from training. On the one hand, completely filtering out shapes in the evaluation categories is technically tricky, and most recent "zero-shot" 3D understanding papers have not done so. On the other hand, we feel that such an evaluation is not necessary. In the 2D CLIP paper, they also don't have such explicit filtering in their "zero-shot" evaluation. As more large-scale 3D datasets become available (e.g., Objaverse-XL has 10M shapes), the training set should cover most object categories. As a result, we believe that a more proper definition of "zero-shot" is the generalization to unseen datasets or evaluation instances. We will follow your suggestions to define that at the beginning clearly.
>
> **Similar linear prob performances on ModelNet40 and ScanObjectNN**
>
> It's hard to figure out the actual reasons for similar linear prob performances on ModelNet40 and ScanObjectNN. We have some hypotheses:
>
> ModelNet40 is relatively easy, and both representations are already good enough. Some confusing shapes mainly cause the remaining error. Many ModelNet40 shapes have very simple geometry and are normalized to a unit scale without color information. As a result, some shapes may have multiple confusing categories (e.g., TV stand, dresser, nightstand). The most effective solution to further reduce the error rate may become including more training shapes and overfitting the subtle bias.
>
> The performance of ScanObjectNN is mainly dominated by the domain gap (e.g., noise from scanned objects). Since both models don't have explicit designs for reducing the domain gap, they share similar performance.
>
> "In-category sample bias" means that when the number of labeled training samples is small, the performance may be greatly affected by the sampled training shapes instead of the quality of shape representation. This is also the reason the few-shot performance varies across different runs, and we report the mean performance over ten runs.
>
> **Additional analyses on “shape representation”**
>
> To analyze the learned shape representations, we added three additional small experiments as suggested by the reviewers:
>
> (a) Shap-latent-based NN retrieval: we utilize shape embedding to retrieve the nearest shapes from the Objaverse dataset. As shown in Figure 1 (see the PDF in the main rebuttal), our retrieved shapes capture more fine-grained geometry details.
>
> (b) Image shape alignment accuracy: we randomly divide the Objaverse-LVIS shapes into batches (196 shapes per batch) and calculate the alignment accuracy between the image (thumbnail of the shapes) and shape embeddings. As shown in Table 3(a), our method achieves a higher alignment accuracy.
>
> (c) We utilize real-world images and bounding boxes in the 2D LVIS dataset to create an image dataset comprising ~3k cropped images from 584 categories. Each resulting image (from a bounding box) is cropped and resized to 224x224 and is associated with a unique category name. For each image, we find the nearest shapes using shape embeddings. If the retrieved top 10/100 shapes of an image contain the corresponding category, we regard the retrieval as a success. We then report the success rate as R@10 and R@100 in Table 3(b). As shown in the table, our method achieves a higher success rate.
>
> **”Limitations are not discussed?”**
>
> We want to clarify that we have discussed the limitations, including limited part-level understanding, sim-to-real gap, etc., in Section 5, “Discussion and Conclusion”. We will move the discussion to a new section, Limitations, in our revision.
>
> **The source of performance improvement**
>
> Yes, we agree that scaling up the 3D backbone may play a critical role in the performance gain (~10 percent). However, other design choices, such as text filtering and enrichment, are also important. For example, without text filtering and enrichment, PointBERT will drop 10.5 percent, as shown in Figure 7 of the main paper.

---

> > ### Comment · Reviewer_quLe · 2023-08-13
> >
> > I thank the authors for their detailed response. Most of my concerns are resolved, and I remain on the positive side.

---

### Official Review · Reviewer_9Jeu · 2023-07-03

**Soundness:** 3 good
**Presentation:** 3 good
**Contribution:** 3 good
**Rating:** 6
**Confidence:** 4

**Summary:**

In this paper, the authors propose a joint learning framework for multi-modal representations among text, image, and point clouds. Specifically, the authors fix the pretrained CLIP language and image encoder, and align the point cloud representation by the proposed multi-modal representation alignment technique. Then, to enlarge the pretraining dataset, the authors introduce text filtering and enrichment technique to annotate point clouds without descriptions. The proposed OpenShape is pretrained on an large-scale ensembled dataset, and the experimental results demonstrate that OpenShape has promising zero-shot classification ability and shape retrieval ability.

**Strengths:**

1. The proposed method fully leverages the large-scale point cloud datasets to align the point cloud backbone with frozen pretrained CLIP encoders, thus obtaining open-world point could recognition and shape retrieval ability.

2. The proposed text filtering and enrichment technique is effective to preprocess large-scale point cloud datasets.

3. The overall writing is polished.

4. The ablation study is extensive and critical.

**Weaknesses:**

My major concern is the fairness in the experiments (i.e., table 2). As the authors stated in section 3.1, the chosen vision-language encoder is OpenCLIP ViT-G-14, which is an extremely large backbone and obtains much better feature representation ability. Therefore, the authors should list the point cloud backbone and pretrained vision-languge encoders concretely in the table, and make fair comparisons if possible.

**Questions:**

See the weakness part.

**Limitations:**

The authors do not state the limitation of this method. Instead, the authors present the future direction of the proposed OpenShape, i.e., part-level information and synthetic-to-realistic domain gap. These directions can be seen as the limitation of this work.

---

> ### Author Rebuttal · Authors · 2023-08-10
>
> Thank you for your insightful comments and valuable suggestions. We will revise our paper based on your feedback. Here are our responses to your comments:
>
> **Vision-language encoder**
>
> Thanks for your suggestion. We will add the details of the point cloud backbone and pretrained vision-language encoders concretely in the table as shown in Table 1 of the PDF file (see main rebuttal).
>
> To evaluate the impact of various pretrained visual-language encoders, we compared CLIP and OpenCLIP encoders on the Objaverse and ShapeNet datasets. As shown in Table 2 of the PDF file, while we do observe performance differences between different visual-language encoders, the gap is insignificant. Note that due to the time limit, we cannot train new models on the Objaverse (or ensembled) dataset at this time. We thus report the models that we have during our previous experiments.

---

> > ### Comment · Reviewer_9Jeu · 2023-08-17
> >
> > Thanks for your response. Most of my concern has been solved.

---

### Official Review · Reviewer_Mx8V · 2023-07-08

**Soundness:** 3 good
**Presentation:** 3 good
**Contribution:** 2 fair
**Rating:** 6
**Confidence:** 5

**Summary:**

This work learns a multi-modal representation among text, images, and point clouds based on a scaling 3D dataset. The results demonstrate remarkable performance in point cloud zero-shot classification, retrieval, and captioning tasks. In order to construct this dataset, the authors combined four commonly used 3D datasets while filtering and enhancing text prompts. Additionally, they introduced an offline hard negative mining strategy to improve the efficiency of the training process.

**Strengths:**

1. This paper collects 0.8M text-image-point cloud triplets from four popular 3D datasets and does text cleaning and enrichment to get high-quality prompts for each object.
2. This work proposes a hard negative mining strategy to improve joint representation training efficiency and performance.
3. The trained representation shows surprising performance on zero/few-shot classification.

**Weaknesses:**

1. The major problem is an insufficient novelty compared with ULIP. The pertaining method, triplets construction and downstream tasks are very similar to ULIP.  The primary distinguishing point of this work is the text prompt cleaning and enrichment flowchart, as well as the offline hard negative mining strategy employed. However, the latter strategy lacks an ablation study to provide supporting evidence.

2. The motivation behind this work is not clearly articulated. While the aim is to develop a scaling-up 3D representation, the purpose of this representation is unclear. Unlike ULIP, which focuses on enhancing current 3D backbones through developing a new pipeline, this work attempts to scale up text-to-3D pairs through text augmentation that is still limited by the insufficient number of 3D objects.

3. There is a lack of exploration into the downstream applications of pre-trained representation to counterparts in 2D. This work is more suitable for submission to the dataset and benchmark track.


**Questions:**

- The main contribution of this work is the datasets ensemble. Will the authors release the collected dataset and pre-trained 3D representation, similar to ULIP?

- How will multiple object combinations and scene models be addressed in the Objaverse dataset? Will such data be ignored? According to Table 2, pre-training on the assembled dataset has a negative impact on the ScanObjectNN dataset, indicating overfitting in the Objaverse dataset and a degradation of the transferability between the simulation and real domains.

- Section 3.4 appears to be an experimental setup and should be placed in the experiments and analysis chapter.

- Lack of related work: Contrastive Language-Image-Point Pretraining from Real-World Point Cloud Data

**Limitations:**

No needed

---

> ### Author Rebuttal · Authors · 2023-08-09
>
> Thank you for your insightful comments and valuable suggestions. We will revise our paper based on your feedback. Here are our responses to your comments:
>
> **The difference between the proposed method and ULIP**
>
> While we follow the multi-modal contrastive learning framework for representation alignment as used in many previous works (not just ULIP), our objective is to learn a useful 3D representation to enable open-world 3D shape understanding by scaling up the pre-training. Here are some main differences:
> ULIP mainly experiments on small-scale datasets (e.g., ShapeNet), while we aim to pretrain on a large-scale ensembled dataset and address the arised challenges. The experiments show that even if ULIP is trained on our ensembled dataset, it’s still less effective than our method.
> ULIP utilizes the original meta information (e.g., model category and name) of 3D dataset for training, while the meta information may be missing or noisy when scaling up the dataset. We thus propose several strategies to filter and enrich text descriptions for the 3D models, which is proved to be effective and critical.
> ULIP utilizes the off-the-shelf 3D backbones with default settings. We explore the influences and scaling laws of 3D backbones when pretraining on a large-scale dataset, which is proven to be critical when learning a scalable representation.
> We propose a hard negative mining module to address the severe class imbalance issue of the ensembled 3D dataset. The module is optional and can bring additional performance gain (as ablated in Table 3). Moreover, it can enable more efficient pretraining. We observe a ~2x speed up when using hard negative mining.
> In addition to the supreme performance over ULIP (both original version and retrained version) on zero/few shot classification, we showcase a lot of cross-modal applications enabled by our powerful representations, which are missing from the ULIP paper. The applications include: (a) image and point cloud input 3D shape retrieval (Figures 1, 8, S1, S2), (b) text input 3D shape retrieval (Figure 9), which supports fine-grained subcategories, attributes and their combinations; (c) point cloud captioning (Figure 10 and S3); (d) point cloud conditioned image generation (Figure 10 and S4). We kindly refer the reviewers to the supplementary material for more exciting results. Note that the ULIP paper only shows some simple cases of image-based retrieval (coarse category-level), and does not demonstrate the capability of open-world understanding and supporting a wide range of downstream applications.
>
> **Purpose of this representation**
>
> Our goal is not to enhance a specific 3D backbone as in ULIP. Instead, we aim to learn a powerful multi-modal representation like in 2D CLIP that could enable open-world 3D shape understanding and directly benefit various downstream applications. As a result, we don’t focus on a specific 3D backbone and explore the scaling law of 3D backbones as well. In the paper, we demonstrate the open-world capability of our method through a wide range of cross modal applications.
>
> **Still limited by the insufficient number of 3D objects**
>
> As more and more large-scale 3D datasets become available (e.g., Objaverse-XL has 10M 3D shapes), we believe that 3D representation learning may no longer be limited by the scale of 3D shapes, and open-world 3D shape understanding is more likely to be realized than ever before. As a result, learning a powerful representation and addressing the challenges arose during the scaling up (the goal of this paper) may become more important and receive more attention. In addition to the large-scale 3D datasets, leveraging and distilling the 2D pre-trained models may also provide clues for this goal.
>
> **Lack of exploration into the downstream applications**
>
> In addition to the supreme performance on zero/few shot classification, we showcase a lot of cross-modal applications enabled by our pre-trained powerful representations. The example applications include: (a) image and point cloud input 3D shape retrieval (Figures 1, 8, S1, S2), (b) text input 3D shape retrieval (Figure 9), which supports fine-grained subcategories, attributes and their combinations; (c) point cloud captioning (Figure 10 and S3); (d) point cloud conditioned image generation (Figure 10 and S4). We kindly refer the reviewers to the supplementary material for more exciting results, which demonstrates that our shape representations encode a broad range of semantic and visual concepts, and enable open-world understanding.
>
> **Code release**
>
> Yes, we will release our code, processed training data, pretrained 3D representations, and demo!
>
> **Scene understanding**
>
> The goal of this paper is single object understanding. However, we believe our work can be a good building block for scene-level 3D understanding, since 3D scene data is much less compared to 3D object data (e.g. 10M shapes). It is interesting to explore how our work can be extended to open-world scene understanding, which we leave as future work.
>
> **Degradation on scanned objects**
>
> Yes, we also observe the performance degradation on the ScanObjectNN dataset mainly due to the sim-to-real domain gap. As mentioned in the “Discussion and Conclusions”, exploring explicit designs (e.g. data augmentation) to reduce the domain gap with scanned objects is challenging but critical and we leave it as future work.
>
> **Lack of related work**
>
> Thanks for pointing this out. We will add the paper in our revision.

---

> > ### Author Response · Authors · 2023-08-11
> > **Fix the format error for the first response**
> >
> > **The difference between the proposed method and ULIP**
> >
> > While we follow the multi-modal contrastive learning framework for representation alignment as used in many previous works (not just ULIP), our objective is to learn a useful 3D representation to enable open-world 3D shape understanding by scaling up the pretraining. Here are some main differences:
> >
> > (1) ULIP mainly experiments on small-scale datasets (e.g., ShapeNet), while we aim to pretrain on a large-scale ensembled dataset and address the arisen challenges. The experiments show that even if ULIP is trained on our ensembled dataset, it’s still less effective than our method.
> >
> > (2) ULIP utilizes the original meta information (e.g., shape category and name) of 3D datasets for training, while the meta information may be missing or noisy when scaling up the dataset. We thus propose several strategies to filter and enrich text descriptions for the 3D models, which is proved to be effective and critical.
> >
> > (3) ULIP utilizes the off-the-shelf 3D backbones with default settings. We explore the influences and scaling laws of 3D backbones when pretraining on a large-scale dataset, which is proven to be critical when learning a scalable representation.
> >
> > (4) We propose a hard negative mining module to address the severe class imbalance issue of the ensembled 3D dataset. The module is optional and can bring additional performance gain (as ablated in Table 3). Moreover, it can enable more efficient pretraining. We observe a ~2x speed up when using hard negative mining.
> >
> > (5) In addition to the supreme performance over ULIP (both the original version and retrained version) on zero/few-shot classification, we showcase a lot of cross-modal applications enabled by our powerful representations, which are missing from the ULIP paper. The applications include:
> >
> >   (a) image and point cloud input 3D shape retrieval (Figures 1, 8, S1, S2),
> >
> >   (b) text input 3D shape retrieval (Figure 9), which supports fine-grained subcategories, attributes, and their combinations;
> >
> >   (c) point cloud captioning (Figure 10 and S3);
> >
> >   (d) point cloud conditioned image generation (Figure 10 and S4).
> >
> > We kindly refer the reviewers to the supplementary material for more exciting results. Note that the ULIP paper only shows some simple cases of image-based retrieval (coarse category level), and does not demonstrate the capability of open-world understanding and supporting a wide range of downstream applications.

---

> > ### Comment · Reviewer_Mx8V · 2023-08-12
> >
> > Thanks for your response. Most of my concern has been solved.

---

### Official Review · Reviewer_Q47N · 2023-07-08

**Soundness:** 3 good
**Presentation:** 3 good
**Contribution:** 3 good
**Rating:** 6
**Confidence:** 5

**Summary:**

OpenShape explores scaling-up strategy for learning joint representations of texts, image, and 3D point clouds. It proposes to construct larger-scale 3D datasets, filter and enrich paired texts for pre-training, which achieves SOTA results on zero-shot 3D classification and retrieval benchmarks.

**Strengths:**

1. The zero-shot performance of OpenShape is impressive, surpassing existing SOTA methods with large margins. This demonstrates the significance of the method.

2. The retrieval visualizations are interesting and clearly illustrate the embedding ability of OpenShape.

3. The authors re-train exiting methods under new settings, which is good for a fair comparison.

**Weaknesses:**

1. OpenShape utilizes GPT-4 for textual-level processing, which however is expensive to access. How about the performance using a more affordable GPT-3 (like PointCLIP V2) ?

2. I'm curious about how OpenShape can be incorporated with non-parametric 3D network Point-NN (or its parametric derivative Point-PN) for zero-shot learning?

Starting from Non-Parametric Networks for 3D Point Cloud Analysis, CVPR 2023

**Questions:**

None

**Limitations:**

Yes

---

> ### Author Rebuttal · Authors · 2023-08-10
>
> Thank you for your insightful comments and valuable suggestions. We will revise our paper based on your feedback. Here are our responses to your comments:
>
> **GPT-4 is expensive to access**
> We also tried GPT-3 in our experiments but found it less effective at reasoning and finishing the filtering task. In our experiments, the average cost of filtering 1000 shapes using GPT-4 is about $0.5, which is still competitive with manual filtering.
>
> **Incorporate with Point-NN or Point-PN?**
> Thanks for pointing this out. Incorporating the parametric version Point-PN into our framework seems possible and interesting. Since Point-PN has only a small number of learnable parameters, it may be more effective when scaling up the model size and enabling large batch sizes. The non-parametric version Point-NN may not be very suitable for our scaled-up experiments since it requires creating a large-scale feature memory for all training shapes, and comparing with such a huge feature memory may be less efficient during inference. Also, it’s not straightforward to enable zero-shot classification (arbitrary texts) using Point-NN, since the point embeddings are required to align with CLIP text embeddings.

---

### Author Rebuttal · Authors · 2023-08-10

Dear Reviewers,

Thank you for dedicating your time to review our paper and for offering insightful feedback. We deeply appreciate your efforts to help enhance the quality of our research. We are also pleased to note that all reviewers were supportive of our work:

(a) Acknowledge our remarkable performance on zero-shot classification (Q47N, Mx8V, 9Jeu, quLe, zuqY) and shape retrieval (Q47N, Mx8V, 9Jeu, quLe).

(b) Praise our extensive and critical analysis and ablation studies, which provide valuable insights to the community and future research along this direction (9Jeu, quLe, zuqY).

(c) Find our applications are interesting and great, which clearly illustrate the embedding ability (Q47N), and the pretrained models can be practically very useful (quLe).

(d) Find our paper well-written and easy to follow (9Jeu, quLe).

---

### Decision · Program_Chairs · 2023-09-21

**Decision:**

Accept (poster)

**Comment:**

This submission received unanimously positive reviews from the reviewers, who liked the formulation and performance. The AC agrees and recommends acceptance.  The authors are encouraged to incorporate the feedback in the camera-ready version.